# Coping with the mental health impact of COVID–19: A study protocol for a multinational longitudinal study on coping and resilience during the COVID-19 pandemic

Insa Backhaus[1‡], Felix Sisenop[2‡], Edvaldo Begotaraj[3], Marija Jevtic[4,5], Simone Marchini[6], Alessandro Morganti[7], Mihail Cristian Pirlog[8], Matej Vinko[9], Milica P. Kusturica[10], Jutta Lindert[2,11]*, the COPERS consortium[¶]

1 Institute of Medical Sociology, Centre for Health and Society (CHS), Heinrich-Heine-University Düsseldorf, Düsseldorf, Germany, 2 Department of Health and Social Work, University of Applied Science Emden/Leer, Emden, Germany, 3 Department of Developmental and Social Psychology, Sapienza University of Rome, Rome, Italy, 4 Faculty of Medicine Novi Sad, University of Novi Sad, Novi Sad, Serbia, 5 Research Centre on Environmental and Occupational Health, School of Public Health, Université Libre de Bruxelles, Brussels, Belgium, 6 Department of Child and Adolescent Psychiatry, Erasme Hospital, Université Libre de Bruxelles, Brussels, Belgium, 7 Department of Architecture, Built Environment and Construction Engineering (DABC), Design & Health Lab, Politecnico di Milano, Milan, Italy, 8 Faculty of Medicine, University of Medicine and Pharmacy of Craiova, Craiova, Romania, 9 National Institute of Public Health, Ljubljana, Slovenia, 10 Faculty of Medicine, University of Novi Sad, Novi Sad, Serbia, 11 WRSC Brandeis University, Waltham, Massachusetts, United States of America

‡ These authors share first authorship on this work.
¶ On behalf of the COPERS consortium as listed in the acknowledgments.
* jutta.lindert@hs-emden-leer.de

**Data Availability Statement:** No datasets were generated or analysed during the current study. All

## Abstract

### Background

Mental health is challenged due to serious life events such as the COVID-19 pandemic and can differ by the level of resilience. National studies on mental health and resilience of individuals and communities during the pandemic provide heterogeneous results and more data on mental health outcomes and resilience trajectories are needed to better understand the impact of the pandemic on mental health in Europe.

### Methods

COPERS (Coping with COVID-19 with Resilience Study) is an observational multinational longitudinal study conducted in eight European countries (Albania, Belgium, Germany, Italy, Lithuania, Romania, Serbia, and Slovenia). Recruitment of participants is based on convenience sampling and data are gathered through an online questionnaire. gathering data on depression, anxiety, stress-related symptoms suicidal ideation and resilience. Resilience is measured with the Brief Resilience Scale and with the Connor-Davidson Resilience Scale. Depression is measured with the Patient Health Questionnaire, Anxiety with the Generalized Anxiety Disorder Scale and stress-related symptoms with the Impact of Event Scale Revised- Suicidal ideation is assessed using item 9 of the PHQ-9. We also consider

relevant data from this study will be made available upon study completion.

**Funding:** Authors received no specific funding for this work.

**Competing interests:** The authors declared that no competing interests exist.

potential determinants and moderating factors for mental health conditions, including socio-demographic characteristics (e.g., age, gender), social environmental factors (e.g., loneliness, social capital) and coping strategies (e.g., Self-efficacy Belief).

## Discussion

To the best of our knowledge, this is the first study to multi-nationally and longitudinally determine mental health outcomes and resilience trajectories in Europe during the COVID-19 pandemic. The results of this study will help to determine mental health conditions during the COVID-19 pandemic across Europe. The findings may benefit pandemic preparedness planning and future evidence-based mental health policies.

## Introduction

With the spread of the COVID-19 pandemic, living on a day-to-day basis has become a challenge for many individuals worldwide. Since the official recognition of the pandemic in March 2020, more than five million people have died due to COVID-19 worldwide (as of December 2021) and at least 262.000.000 people have been infected [1]. In addition to the physical health impact of the pandemic, impacts on mental health, such as a higher risk of anxiety, elevated suicidal ideation and increased worries, have been reported in China [2], the USA [3] and Europe [4, 5]. Stressors put forward to negatively affect mental health include social isolation, loneliness, fear of contagion, interruption of healthcare services, loss of income, and loss of employment [5]. During the course of the pandemic, concerns have also been raised about the unequal distribution of mental health problems. While some groups seem to be more resilient, other groups are more vulnerable to the psychological impacts of the pandemic [6, 7], including children and adolescents [8, 9], people facing job insecurity [8] and people with pre-existing mental disorders [10].

Resilience has become an important concept in explaining differences in mental health when experiencing stressful life events [11, 12]. Essentially, resilience describes the ability to withstand setbacks, adapt positively, and bounce back from adverse life events such as disasters [13]. Scientific evidence shows that individuals with greater resilience respond more adequately to both natural and man-made disasters than individuals with lower resilience [14]. While research about the importance of resilience for mental health during the pandemic is emerging, it remains unclear, which factors promote or decrease resilience in times of crisis [15–17]. Researchers from the USA and China, for instance, found that individual and family resilience are associated with mental health [16]. However, the authors did not investigate factors that may contribute to resilience. Currently, research examining determinants of resilience is only available from before the COVID-19 pandemic. A pre-pandemic study from Japan, for instance, points towards an association between social capital and resilience in Japan [18]. However, determinants of resilience may be situation-dependent and vary between countries and different population groups. In addition, resilience has been suggested to change over time as a result of personal development or one's interaction with the environment [19, 20], but at present, most research on resilience uses cross-sectional data that does not allow the investigation of change over time. Moreover, current measures to contain the spread of COVID-19, such as lockdown, quarantine, and contact restriction, may complicate matters even further as they have changed people's social behavior and social activities overnight. Factors such as feeling lonely and depressed during the lockdown as well as enhanced anxiety and

suicidal ideation, may negatively affect resilience [21]. Very recent research from Panzeri and colleagues (2021), showed that COVID-19-anxiety, intolerance of uncertainty and loneliness hindered resilience [21]. Thus, a range of factors may promote or hinder resilience and consequently affect mental health.

Given that to date, only a very few studies assess factors that facilitate or hinder resilience, COPERS (Coping with COVID-19 with Resilience study) was developed. COPERS is a multinational longitudinal study among eight European countries that aim to investigate how mental-health outcomes vary between countries and to identify determinants of resilience and resilience trajectories over time. The findings will be important, as understanding, which factors foster resilience and mental health can help to inform effective health and social policies.

## Objective, aims and hypotheses

The specific aims of COPERS are to (1) investigate the prevalence rate of depression, suicidal ideation and stress related symptoms over the course of the pandemic and whether the prevalence rate varies between European countries; (2) to identify factors that can facilitate or hinder resilience and to identify resilience trajectories; and (3) to determine how resilience is linked to mental health during the COVID-19 pandemic. Our main hypotheses are:

1. We hypothesize that prevalence rate of depression, suicidal ideation and stress-related symptoms such as anxiety, increase over the course of the pandemic.

2. We hypothesize that prevalence rates of depression, suicidal ideation and stress-related symptoms such as anxiety, vary between countries.

3. We hypothesize that factors such as one's socioeconomic position, social support, social capital, loneliness, alcohol consumption, and exposure to the COVID-19 pandemic are associated with resilience trajectories.

4. We hypothesize that participants of a higher socioeconomic position and with a greater level of social capital and social support are less likely to experience poor mental health outcomes (e.g., depression, suicidal ideation and stress-related symptoms), whereas participants of a lower socioeconomic position and with greater levels of loneliness, higher levels of alcohol consumption and exposure to a greater pandemic burden are more likely to experience poor mental health outcomes (e.g., depression, suicidal ideation and stress-related symptoms).

## Methods

COPERS is a non-funded, multinational longitudinal study and is run as a European collaborative research project composed by members of the EUPHA (European Public Health Association) public mental health group. It is led by researchers based at the University of Applied Sciences Emden/Leer in Germany, who have previously designed and delivered several projects across the European Union.

### Study population

The sample consists of participants aged 18 years and older from Albania, Belgium, Germany, Italy, Lithuania, Romania, Serbia and Slovenia. To be eligible for the study, the potential participants must satisfy the following criteria:

- Person must be aged 18 years or older

- Person must have access to internet

- Person must be able to provide informed consent online using the study website

- Person must be proficient in one of the following languages: Albanian, Dutch, English, French, German, Italian, Lithuanian, Romanian, Slovenian, and Serbian.

## Recruitment and study procedure

In all countries, recruitment and sampling approaches are based on an opportunistic, convenience–pandemic-specific sampling approach. In the present context of a pandemic, opportunistic sampling means using snowball sampling and following a purposive sampling approach using a diversity of pandemic-appropriate recruitment strategies. In the present study, the recruitment strategies include recruitment through homepages, professional associations, homepages of the participating institutions, social media and its services, press and media releases, and direct personal contacts. A complete list detailing the sampling strategies in each country is provided in the supplemental materials.

Prior to participating in the study, all potential participants are asked to complete an informed consent question embedded on the first page of the online survey. In case the respondents agree to participate in the study (i.e., entered "Yes" to the consent question), they can fill out the survey and a further invitation will be sent to collect data on the second and third wave. Respondents who disagree to participate in the study (i.e., entered "No" to the consent question) are directly transferred to the end of the survey. Participation is voluntary and can be withdrawn at any time.

## Data collection

For the proposed study, a self-administered online survey will be applied. The survey will be comprised of four parts. The first part will collect sociodemographic information (e.g., age, gender, degree of education). The second part will investigate mental health (e.g., depression) and the third part will gather information on resilience trajectories. The final part will gather information on possible determinants of resilience, mental health and moderating factors such as social capital, general self-efficacy, alcohol use and loneliness. The survey is predominantly based on previously standardized and validated instruments and will include only a few self-developed COVID-19 specific questions. The choice of the instruments is based on appropriateness for the context, reliability, and validity.

## Primary outcomes

**Resilience.** Resilience is defined as the ability to adapt to a changing situation and changing circumstances in the physical and social environment [22]. Accordingly, resilience will be assessed with two measures, using the Brief Resilience Scale (BRS) and the Connor-Davidson Resilience Scale (CD-RISC) [23]. The BRS was created to measure the ability to bounce back or recover from stress and is a valid and reliable tool for assessing resilience [24]. The BRS uses six items to assess resilience, based on a Likert agreement scale ranging from 1 (strongly disagree) to 5 (strongly agree) [24]. The CD-RISC is one of the most widely used scales to measure psychological resilience. It includes 25 items rated on a five-point Likert scale and range from 0 (not true at all) to 4 (true nearly all the time). Possible scores range from 0 to 100. The CD-RISC has been shown to predict both physical and mental health-related quality of life and to be of good reliability and validity [23]. Both the BRS and the CD-RISC are available in the respective survey languages applied in this study.

**Depression and suicidal ideation.** The primary mental health outcome is depression, measured with the Patient Health Questionnaire (PHQ-9), a multipurpose instrument for screening, diagnosing, monitoring, and measuring the severity of depression [25]. The PHQ-9 is a depression assessment tool, which scores each of the 9 DSM-IV criteria from 0 (not at all) to 3 (nearly every day). PHQ-9 cut-off scores of 5, 10, 15, and 20 represent mild, moderate, moderately severe, and severe depression, respectively. The instrument yields scores ranging from 0 to 27, with higher scores indicating more severe depressive symptoms [26]. The assessment of suicidal thoughts will be based on item 9 of the PHQ-9. Item 9 of the PHQ-9 asks respondents to indicate if they have had thoughts of hurting themselves in some way or thoughts that they would be better off dead. Item 9 of the PHQ-9 is a robust predictor of suicide attempts [27]. Generally, the PHQ-9 has been shown to be a valid and reliable measurement tool for depression across European countries [25, 28] and has been translated into Dutch, French, German, Italian, Lithuanian, Romanian, and Serbian [29]. To be able to use the PHQ-9 in Albania and Slovenia; the tool was translated using the forward-backward translation approach.

**Stress symptoms.** The Impact of Event Scale Revised (IES-R) will be used for measuring stress [30]. The IES-R is a 22-item self-reported measure that assesses subjective distress caused by traumatic events. The IES-R has been translated into many languages including Germany, French, and Italian. A team of researchers translated the IES-R into Albania; Dutch, Lithuanian, Romanian, Slovenia and Serbian using the forward-backward translation approach.

**Anxiety.** Symptoms of anxiety will be measured using the generalized anxiety disorder 7 (GAD-7). The GAD-7 is a seven-item self-administered scale with a total score between 0 to 21. Higher scores indicate higher severity of anxiety (0–4 = minimal, 5–9 = mild, 10–14 = moderate and 15–21 = severe) [31]. The GAD-7 has been translated into Dutch, French, German, Italian, Lithuanian and Romanian [29]. Given the GAD-7 is not available in Albanian, Slovenian and Serbian, the forward-backward translation approach was applied to translate the tool into the respective languages.

## Determinants of resilience, mental health and stress

A variety of determinants have been linked to resilience, mental health and stress, and have been reported to contribute to the association between stressful life events and mental health. These include, among others, sociodemographic factors, social capital, anxiety and loneliness. These determinants are also measured in the present study, as described below.

**Sociodemographic data.** The sociodemographic data will be collected on age, gender, living environment during the pandemic, educational level, occupational status, income, having children and marital status.

**Social capital.** Social capital that has been defined as "connections among individual social networks and the norms of reciprocity and trustworthiness that arise from them" [32], has been linked to mental health [33]. A growing body of literature suggests that in times of crisis, higher levels of social capital can enhance an individual's ability to respond and recover from such crises, thus positively influencing resilience and mental health. The present study will focus particularly on cognitive social capital, which will be assessed using items drawn from the World Bank Integrated Questionnaire to Measure Social Capital (IQ-SC), a psychometric validated instrument [34].

**Loneliness.** The revised UCLA (University of California, Los Angeles) Loneliness Scale was used to measure perceived loneliness. The scale includes 20 items to measure subjective feelings of loneliness and has been shown to have good validity and reliability [35]. Participants are asked to rate items on a scale from 0 (I never feel this way) to 3 (I often feel this way) [35].

The tool has been validated in many different countries and translations in various languages exist [36, 37].

**Alcohol consumption.** Alcohol consumptions will be explored using the Alcohol Use Disorders Identification Test Concise (AUDIT-C) [38]. The AUDIT-C consists of the first three items of the full AUDIT (Alcohol Use Disorder Identification Test) and measures the typical frequency of alcohol consumption, the usual quantity per occasion, and the frequency of heavy episodic drinking [38]. The AUDIT-C has been found to be valid and reliable across various settings and different racial/ethnic groups and has been translated into various languages [39, 40].

**COVID-19 burden.** To explore specific COVID-19 pandemic-related factors, a set of self-developed COVID-19 pandemic-specific items (drawing from our expertise) will be included in the survey. The items will gather information on contamination anxiety, compliance with public health measures and separation of loved ones.

## Statistical methods

Descriptive and inferential statistics will be applied to confirm or reject our hypotheses. To investigate prevalence rates of depression, suicidal ideation and stress-related symptoms such as anxiety as well as differences in prevalence rates between countries, descriptive statistics will be applied. To assess the association between social capital, loneliness and resilience as well as between resilience and mental health outcomes multilevel logistic regression analyses will be conducted. To identify moderating and mediating factors moderation and mediation analyses will be applied. Sensitivity analyses (excluding e.g., high-income countries) will contribute to a better understanding of the generalizability of our findings. More specifically, every measurement point, starting from the baseline data, will be analyzed cross-sectionally and longitudinally concerning the objectives of COPERS. To examine resilience trajectories linear, logistic and multinomial logistic regression analyses, will be conducted. Furthermore, latent growth curve analyses in the framework of structural equation modelling analyses will be conducted when testing which factors predict changes in resiliency and mental health trajectories. Furthermore, to simultaneously control for many variables that potentially confound the relationship between mental health and resilience, propensity score matching will be applied [41]. Propensity score matching offers the advantage of ensuring the baseline distribution of confounders between groups, which can increase group comparability. In addition, a multilevel analytical approach will be used to determine the various extrinsic and intrinsic influences on health and development.

Furthermore, sex, gender, age, and diversity are considered at all stages of the planned project. Wherever possible, a respective subgroup analysis will be performed. For instance, if possible, regression models will be run separately for different age groups. Data will be collected at different time points and loss of follow-up, drop-out and incompleteness are to be expected. The survey will be distributed at different time points. A sensitivity analysis will be used to assess how different time points may have affected the outcomes. All analyses will be performed in IBM SPSS version 26.0 and Stata version 15.0.

## Methods to handle protocol non-adherence and any statistical methods to handle missing data

Descriptive statistics will be generated for two categorical variables: (1) full sample analyzed and (2) missing data. Missing data may lead to bias and loss of information in epidemiological research. If more than 15% of data are missing, multiple data imputation will be performed. In

order to assess the potential risk of bias, we will perform a sensitivity analysis using the imputed data set.

## General ethical aspects

This study will adhere in all stages of the project to the ethical principles of informed consent, voluntary participation, do no harm, confidentiality, anonymity and only assess relevant components. In practice, the study will (a) obtain informed consent from potential research participants; (b) minimize the risk of harm to participants; (c) protect their anonymity and confidentiality; (d) avoid using deceptive practices; and (e) give participants the right to withdraw from the research. This study will follow the Nuremberg Code (1947) and the Helsinki code (1964) at all stages of the research project. Our research does not include vulnerable populations (e.g., prisoners, institutionalized persons). The Ethics Committee of the University of Emden/Leer approved the COPERS and all Institutional Review Boards (IRB) of the participating sites. The survey will involve multiple sensitive questions, including ones about depressive symptoms and suicidal ideation. Therefore, answering may cause discomfort for participants or distress in respondents with pre-existing emotional vulnerabilities. Participants will be provided with information about counselling centers. Furthermore, participants do not have to answer questions they feel uncomfortable with and can stop compilation at any time.

Any important modification to the protocol that may impact on the conduct of the study, including changes of study objectives, study design, study procedures, or significant administrative aspects, will be communicated to the ethical committees of all participating project partners.

## Study status and timeline

COPERS is currently on-going, and the timeline and major milestones are shown in Fig 1. Participant enrollment started in 2020 and data collection for Wave 1 commenced in August 2020. Data collection for Wave 2 is planned for the summer 2021 and data collection for Wave 3 is expected to start at the beginning of 2022.

## Discussion

To the best of our knowledge, there are no comprehensive longitudinal studies on resilience and mental health during the COVID-19 pandemic in the European region so far. COPERS has been designed as the first multinational longitudinal prospective study in eight countries

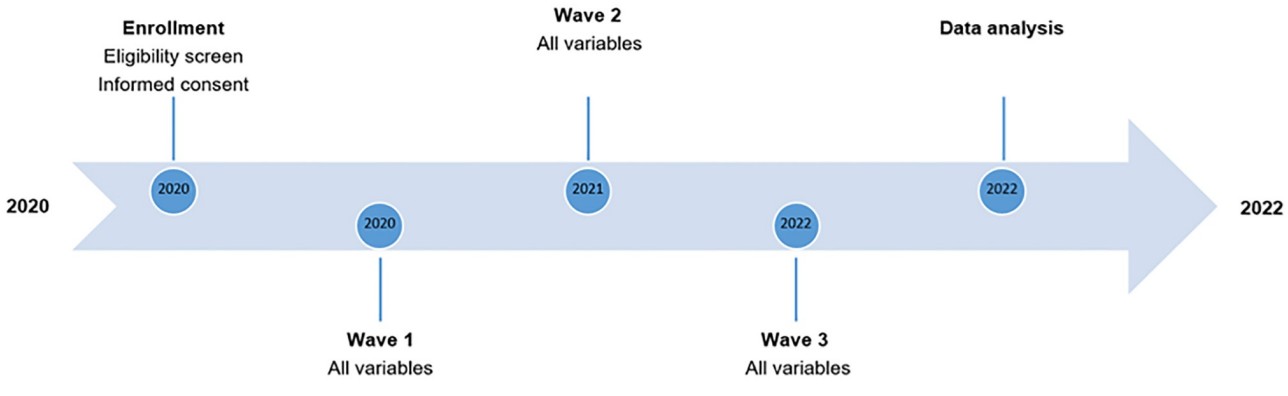

**Fig 1. COPERS timeline.**

in Europe, assessing mental health and resilience trajectories during the COVID-19 pandemic. Therefore, it makes two important contributions: First, COPERS is the first study to document resilience trajectories and mental health outcomes during the COVID-19 pandemic in a multi-national sample from middle-income to high-income countries across Europe. In addition, COPERS is among the first studies to include Eastern European countries such as Albania, Serbia and Lithuania, for which very limited data about mental health and resilience are available. Second, with its longitudinal and multilevel design, it will provide reliable information on changes in resilience trajectories and mental health outcomes and contextual and macro level factors possibly influencing mental health and resilience of the European population during the pandemic. A longitudinal and multilevel design is particularly important because, to date, the majority of studies that have examined resilience applied a cross-sectional design and were unable to investigate the ways in which resilience changes and interacts with events over time.

## Implications for research and public health professionals

The COVID-19 pandemic has hit the world in unprecedented ways and has led to a stark increase in mental ill-health. The results of this study will be of importance for researchers and public health professional for at least two reasons: First, the COVID-19 pandemic is unlikely the last crisis and pandemic. To promote as well as to prevent declining mental health in times of a crisis, it is therefore of great importance to identify factors that can foster resilience and promote mental health ahead of a crisis. By examining resilience trajectories and factors associated with mental health, the results of this study offers novel knowledge of modifiable factors can be helpful for decision-making of public health professional, health professionals, and policymakers to prepare for other disease outbreaks and pandemics, where containment measures are necessary. Second, the study examines the underpinnings of resilience and the findings of this study may help to build resilient communities and to inform pandemic preparedness planning as well as health and social policies.

**Strengths and limitations.** Despite the valuable information provided by COPERS, some limitations need to be mentioned. First, the findings will be based on a convenience sample. Therefore, it is possible that the sample is not representative of an entire country or region, limiting the generalizability of the results of the survey to the populations as a whole. Second, given that the study is a web-based survey, it is possible that people who do not have access to the internet or do not know how to use digital technology will be underrepresented (e.g., elderly). Consequently, selection bias is possible. Third, mental health outcomes are assessed using self-reported measures and therefore information or recall bias cannot be ruled out. Despite the good reliability and validity of the PHQ-9 and the GAD-7, the involvement of an assessment by a clinician is always preferable when assessing mental health.

## Conclusion

To conclude, COPERS is a unique study and one of the largest study of its kind in Europe to collect longitudinal data on resilience and mental health during the COVID-19 pandemic. Results of the study will improve the understanding of the ways in which resilience and mental health are influenced by individual, situational and contextual factors. The findings of this study can help to build resilient communities and to inform pandemic preparedness planning as well as health and social policies.

## Supporting information

**S1 Table. Instruments used in COPERS.**
(DOCX)

**S1 File. Genehmigung ethikantrag COPERS.**
(PDF)

**S2 File. P2021-142- accord CE.**
(PDF)

**S3 File. POLIMI IRB approval.**
(PDF)

**S4 File. UMFCV IRB COPERS.**
(PDF)

## Acknowledgments

The COPERS consortium consists of Insa Backhaus, Edvaldo Begotaraj, Stefano Capolongo, Mauro G. Carta, Marija Jakubauskiene, Marija Jevtic, Milica P. Kusturica, Jutta Lindert, Alessandro Morganti, Mihail Cristian Pirlog, Andrea Rebecchi, Felix Sisenop, and Matej Vinko.

### For the COPERS consortium

Stefano Capolongo (Department of Architecture, Built Environment and Construction Engineering (DABC), Design and Health Lab, Politecnico di Milano, Milan, Italy), Marija Jakubauskiene (Faculty of Medicine, Vilnius University, Vilnius, Lithuania), Vladimir Nakov (Department of Mental Health, National Center of Public Health and Analyses, Sofia, Bulgaria),

## Author Contributions

**Conceptualization:** Jutta Lindert.

**Data curation:** Felix Sisenop, Edvaldo Begotaraj, Jutta Lindert.

**Formal analysis:** Insa Backhaus, Felix Sisenop, Simone Marchini, Mihail Cristian Pirlog, Jutta Lindert.

**Investigation:** Jutta Lindert.

**Methodology:** Insa Backhaus, Felix Sisenop, Marija Jevtic, Jutta Lindert.

**Supervision:** Jutta Lindert.

**Validation:** Edvaldo Begotaraj, Simone Marchini, Alessandro Morganti.

**Visualization:** Alessandro Morganti, Jutta Lindert.

**Writing – original draft:** Insa Backhaus, Jutta Lindert.

**Writing – review & editing:** Edvaldo Begotaraj, Marija Jevtic, Simone Marchini, Alessandro Morganti, Mihail Cristian Pirlog, Matej Vinko, Milica P. Kusturica.

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
