## [Decision Letter · Decision Letter 0]

21 Mar 2023

PONE-D-23-00090Coping with the mental health impact of COVID–19: a study protocol for a multinational longitudinal study on coping and resilience during the COVID-19 pandemicPLOS ONE

Dear Dr. Lindert,

Thank you for submitting your manuscript to PLOS ONE. After careful consideration, we feel that it has merit but does not fully meet PLOS ONE’s publication criteria as it currently stands. Therefore, we invite you to submit a revised version of the manuscript that addresses the points raised during the review process.

We look forward to receiving your revised manuscript.

Kind regards,

Chalachew Kassaw Demoze

Academic Editor

PLOS ONE

Journal Requirements:

"The authors declared that no competing interests exist."

4. Please upload a copy of Figure 1, to which you refer in your text on page 12. If the figure is no longer to be included as part of the submission please remove all reference to it within the text.

Reviewers' comments:

Reviewer's Responses to Questions

**Comments to the Author**

1. Does the manuscript provide a valid rationale for the proposed study, with clearly identified and justified research questions?

Reviewer #1: Yes

Reviewer #2: Partly

2. Is the protocol technically sound and planned in a manner that will lead to a meaningful outcome and allow testing the stated hypotheses?

Reviewer #1: Partly

Reviewer #2: Partly

3. Is the methodology feasible and described in sufficient detail to allow the work to be replicable?

Reviewer #1: No

Reviewer #2: Yes

4. Have the authors described where all data underlying the findings will be made available when the study is complete?

Reviewer #1: Yes

Reviewer #2: Yes

5. Is the manuscript presented in an intelligible fashion and written in standard English?

Reviewer #1: No

Reviewer #2: Yes

6. Review Comments to the Author

You may also provide optional suggestions and comments to authors that they might find helpful in planning their study.

Reviewer #1: 1. It would be sounded for the paper to add an important components on the Abstract part as it is a very readable section.

2. While the paper to be sounded Grammar should be corrected especially tense change

3. Sample size calculation and sample size is not clear.

4. Mental health is very broad, so the author be specific from the variables of the studies; For example, the determinant of resilience, mental health and stress, as author put it.

5. The reasons why the study fails to address the finding relate to previous researches is not clear

Reviewer #2: The researchers conducted Coping with the mental health impact of COVID–19: a study protocol for a multinational longitudinal study on coping and resilience during the COVID-19 pandemic. I find that the the introduction is clear, I have a few remarks to improve the manuscript:

• Avoid subjective verbs We, I, Our in the document rather use the study, this study…

• Your hypotheses are not clearly written especially the third and fourth hypothesis, try to be more specific.

• Remove this sentence from inclusion criteria “Persons are not eligible to participate if they do not meet the inclusion criteria”

• Be care on some technical terms between multiple and multivariable terms. Re-correct it

• Add more important implications for health (individual and/or public), as well as health care, policy and for other researchers

• Why you impose language limits,

• Try to answer all of the objectives in clear manner since it is important?

• Make your discussion more strong.

• On conclusion part it is expected to an

• Unify the references

7. PLOS authors have the option to publish the peer review history of their article (what does this mean?). If published, this will include your full peer review and any attached files.

Reviewer #1: No

Reviewer #2: **Yes: **Tilahun Bete

While revising your submission, please upload your figure files to the Preflight Analysis and Conversion Engine (PACE) digital diagnostic tool, https://pacev2.apexcovantage.com/. PACE helps ensure that figures meet PLOS requirements. To use PACE, you must first register as a user. Registration is free. Then, login and navigate to the UPLOAD tab, where you will find detailed instructions on how to use the tool. If you encounter any issues or have any questions when using PACE, please email PLOS at figures@plos.org. Please note that Supporting Information files do not need this step.<quillbot-extension-portal></quillbot-extension-portal>

---

## [Author Response · Author response to Decision Letter 0]

21 Apr 2023

Response letter

Dear Editorial Team and Reviewers,

on behalf of all the co-authors, I thank you very much for your interest in our work and for giving us the chance to revise the manuscript once more. You have brought up some very good points that will surely improve our manuscript. We have tried to do our best to respond to the points raised and we hope it will now be suitable for publication in the Plose One. Please find below your comments (in bold) and our replies.

The researchers conducted Coping with the mental health impact of COVID–19: a study protocol for a multinational longitudinal study on coping and resilience during the COVID-19 pandemic. I find that the introduction is clear. I have a few remarks to improve the manuscript:

Author response: Thank you very much for your comments. We have tried to improve our manuscript further.

Avoid subjective verbs We, I, Our in the document rather use the study, this study…

Author response: Thank you for this comment. We have now avoided subjective verbs.

Your hypotheses are not clearly written especially the third and fourth hypothesis, try to be more specific. 

Author response: Thank you for this impotant comments. We have rephrased hypotheses 3 and 4. It reads as follows:

3. We hypothesize that factors such as one’s socioeconomic position, social support, social capital, loneliness, alcohol consumption, and exposure to the COVID-19 pandemic are associated with resilience trajectories.

4. We hypothesize that participants of a higher socioeconomic position and with a greater level of social capital and social support are less likely to experience poor mental health outcomes (e.g., depression, suicidal ideation and stress-related symptoms), whereas participants of a lower socioeconomic position and with greater levels of loneliness, higher levels of alcohol consumption and exposure to a greater pandemic burden are more likely to experience poor mental health outcomes (e.g., depression, suicidal ideation and stress-related symptoms).

Remove this sentence from inclusion criteria “Persons are not eligible to participate if they do not meet the inclusion criteria”

Author response: We have deleted this sentence from the manuscript. 

Be careful on some technical terms between multiple and multivariable terms. Re-correct it

Author response: We have reviewed the methods section and corrected it where appropriate.

Add more important implications for health (individual and/or public), as well as health care, policy and for other researchers

Author response: We have revised the section for implications for health (individual and/or public), as well as health care, policy and for other researchers. To highlight the importance and the implications of the study we wrote an entire paragraph and this is now a single section in the discussion.

Why you impose language limits.

Author response: We restrict the language to the language of the participating countries and the language skills of collaborating researchers. This is a non-funded research project and providing the survey in additional languages is out of scope. 

Try to answer all of the objectives in clear manner since it is important?

 Author response: We have re-worded the objective to one so it is more in line with hypothesis one. We will try our best to answer all objectives through a careful data analysis once the study is completed.

Make your discussion more strong.

Author response: We have added the implications for health (individual and/or public), as well as health care, policy and for other researchers to the discussion.

On conclusion part it is expected to an

Author response: Please excuse us, but we do not fully understand this comment. We have done a careful English review. 

Unify the references 

Author response: We are sorry but we cannot unify the references.

---

## [Editor Report · Decision Letter 1]

2 May 2023

Coping with the mental health impact of COVID–19: a study protocol for a multinational longitudinal study on coping and resilience during the COVID-19 pandemic

PONE-D-23-00090R1

Dear Dr. Lindert,

We’re pleased to inform you that your manuscript has been judged scientifically suitable for publication and will be formally accepted for publication once it meets all outstanding technical requirements.

Kind regards,

Chalachew Kassaw Demoze

Academic Editor

PLOS ONE

Additional Editor Comments (optional):

It is better to include a conclusion part on the abstract section of the manuscript

It is better to re-write a manuscript with better English grammar

Reviewers' comments:

<quillbot-extension-portal></quillbot-extension-portal>

---

## [Editor Report · Acceptance letter]

9 May 2023

PONE-D-23-00090R1 

Coping with the mental health impact of COVID–19: a study protocol for a multinational longitudinal study on coping and resilience during the COVID-19 pandemic 

Dear Dr. Lindert:

I'm pleased to inform you that your manuscript has been deemed suitable for publication in PLOS ONE. Congratulations! Your manuscript is now with our production department. 

Kind regards, 

on behalf of

Dr. Chalachew Kassaw Demoze 

Academic Editor

PLOS ONE